# Effect of Season and Social Environment on Semen Quality and Endocrine Profiles of Three Endangered Ungulates (*Gazella cuvieri*, *G. dorcas* and *Nanger dama*)

**DOI:** 10.3390/ani11030901

**Published:** 2021-03-22

**Authors:** Lucía Arregui, José Julián Garde, Ana Josefa Soler, Gerardo Espeso, Eduardo R. S. Roldan

**Affiliations:** 1Reproductive Biology and Ecology Group, Museo Nacional de Ciencias Naturales (CSIC), 28006 Madrid, Spain; roldane@mncn.csic.es; 2SaBio IREC (CSIC-UCLM-JCCM), ETSIAM, 02071 Albacete, Spain; Julian.garde@uclm.es (J.J.G.); anajosefa.soler@uclm.es (A.J.S.); 3Estación Experimental de Zonas Áridas (CSIC), 04001 Almeria, Spain; gerardo@eeza.csic.es

**Keywords:** seasonality, housing, sperm parameters, testosterone, cortisol, gazelle

## Abstract

**Simple Summary:**

Good-quality sperm samples are needed for the development and implementation of sperm cryopreservation, in vitro fertilization and artificial insemination. These reproductive biotechnologies play an important role in the conservation and management of domestic and wild species. The aim of this study was to explore the effect of seasonality and social environment on sperm quality in three endangered gazelles: Cuvier’s, dorcas and Mohor gazelles. Periods of better sperm quality were related with higher conception rates in Cuvier’s and Mohor gazelles but not in dorcas. Cuvier’s gazelle showed higher sperm quantity in April and Mohor gazelle in April and August and correlated with environmental data. In dorcas gazelle, a drop in sperm quality was observed in October. Housing conditions did not affect sperm quality in Cuvier’s and Mohor gazelles, whereas dorcas males housed with females showed lower semen quality than males kept alone or with males. Considering these results could improve the success of reproductive biotechnologies in these three species.

**Abstract:**

Knowledge of factors affecting semen quality could be of great importance for the collection and preservation of semen from threatened animals. To assess the effect of seasonality, sperm parameters and testosterone levels were examined throughout the year and compared with the distribution of conceptions. Cuvier’s gazelle showed higher sperm quantity in April, coinciding with one peak of conceptions. In dorcas gazelle, sperm parameters showed a drop in October. However, percentage of conceptions increased during that month. In Mohor gazelle, sperm quality was best in April and August, in agreement with higher conception rates and high testosterone levels. Percentage of conceptions was correlated with photoperiod and rainfall in Cuvier’s gazelle and with temperature in Mohor gazelle. To assess the effect of social environment, semen quality, testosterone and cortisol levels were quantified in males housed alone, in bachelor groups or with females. No differences were seen in Cuvier’s and Mohor gazelles’ semen traits, whereas dorcas males housed with females showed lower semen quality than males kept alone or with other males. Overall, ejaculate quality is influenced by seasonal factors in the three gazelle species, while social factors only appear to affect that of dorcas gazelle.

## 1. Introduction

Research in reproductive physiology is a fundamental area in the biology of conservation. Conservation programs that aim to maintain genetic diversity through the use of reproductive biotechnologies require basic information on reproductive biology and how it is influenced by environmental and social factors. Variation in sperm quality could preclude the success of reproductive biotechnologies such as in vitro fertilization (IVF) or sperm cryopreservation.

Between 1971 and 1975, the “Estación Experimental de Zonas Áridas” (EEZA-CSIC) located in the south of Spain started captive breeding programs for three endangered ungulates whose natural populations are still nowadays decreasing: Cuvier’s gazelle (*Gazella cuvieri*), dorcas gazelle (*G. dorcas neglecta*) and Mohor gazelle (*Nanger dama mhorr*). Currently, Cuvier’s and dorcas gazelles are regarded as “Vulnerable” by the Red List of Threatened Species [1,2]. *Nanger dama* is categorized as “Critically Endangered”, with a population of less than 250 mature individuals [3], and the subspecies *N. dama mhorr* is considered to be extinct in the wild since 1968 [4]. Founding populations have reproduced successfully and animals have been translocated to zoos in Europe (also Spain) and the USA [5,6,7,8].

Births of offspring in these species have been showed to vary throughout the year in captive and wild populations. At the EEZA, Cuvier’s gazelle shows a peak of parturitions in spring and a less pronounced one in autumn [6,9,10], and in the wild, births have been described to occur in spring [4]. Variations have been described in the annual distribution of parturitions in dorcas gazelle kept in captivity [5]. This species is reported to be a seasonal breeder in its natural habitat presenting one (spring) or two (spring and autumn) main periods of births depending on the area [11,12,13]. Births of dama gazelle in Western Sahara have been reported to take place in February and March [12]. In captivity, births of Mohor gazelle have been recorded throughout the year at the EEZA, although peaks have been observed in March–April and September [14]. Similarly, Addra gazelle (*N. dama ruficollis*) kept in zoos in USA showed calves born in each month of the year but birth frequency was higher between July and September [15]. Multiple factors could be modulating the annual distributions of births. Considering the male effect, concentration of births could be due to a better sperm quality due to changes in testicular function or epididymal maturation during certain periods.

Seasonal changes in testicular function vary among species and can range from complete arrest of spermatogenesis during part of the year to no effect on sperm production [16], and this could be one of the factors that may explain the seasonal patterns of offspring births. Among seasonal species, offspring are born during the time of the year when food resources are most abundant. This implies that mating only occurs during a restricted period of time, “the breeding season” [17]. This seasonality is controlled by photoperiod in temperate and polar habitats [18]. However, in arid and tropical habitats, the photoperiod varies minimally, and rainfall is the major factor affecting the availability of food [18].

Seasonality of male reproduction has been studied in some wild bovids, revealing both species-specific differences and variations between captive and free-living populations [19,20,21,22,23]. In addition, many bovids show variation of semen quality, testicular function and/or testosterone throughout the year [16,19,21,24,25,26,27,28,29]. Sperm parameters have been characterized for these three species of gazelle kept at EEZA [30,31]. Also, one study in dorcas gazelle developed in 1981 showed no seasonal variations in seminal characteristics and testosterone levels when observed after less than ten years in captivity in the United States [20]. However, the effect of seasonality on sperm quality in Cuvier’s and Mohor gazelles, and the relation between the peak of conceptions, sperm quality and testosterone in these three species, have never been analyzed.

In addition, social conditions have been shown to influence reproductive physiology in male mammals. For instance, subordinate male mice show lower sperm motility than dominants when groups of males are housed together [32], and spermatogenesis is stimulated in dominant male mice when female odor is presented [33]. In social naked mole-rats, lower sperm number is found in non-breeding males when compared with breeding ones [34]. Meadow voles adjust their sperm morphometry and sperm investment respectively, according to the condition of the social environment [35]. Lambs in a lower level of the social rank reach sexual maturity later than high-ranked individuals [36]. Hence, sharing the enclosure with other males or females could affect ejaculate quality.

Therefore, differences in sperm parameters throughout the year and among adult males housed in bachelor groups, individually or in breeding herds, have not been examined in these species and it is possible that theses environmental and/or social factors could be modulating sperm quality, affecting reproductive biotechnologies, such as the success of sperm cryopreservation. Sperm sensitivity to cryopreservation has been shown to be affected by season in bovids [37,38]. Protocols for semen cryopreservation have been developed with varying success in these gazelles [39,40,41], but firstly, the effect of seasonality and social life on fresh sperm quality need to be assessed.

The objectives of this study were (1) to examine circannual changes in conception frequency in three endangered gazelle species kept in captivity, (2) to analyze variations in semen quality and testosterone levels throughout the year and (3) to assess ejaculate traits, testosterone and cortisol levels in males in different housing conditions.

## 2. Materials and Methods

### 2.1. Animal Maintenance and Environment

Animals used in the present study were maintained by the EEZA (CSIC) at the Parque de Rescate de la Fauna Sahariana (PRFS) located in Almeria in the south of Spain (latitude 36°50′ N and longitude 2°28′ W). Mean monthly temperatures, rainfall and light hours at the time of sampling are shown in Figure 1 (Plataforma Solar de Almeria, personal communication).

Animals were fed daily in the morning with barley, granulated feed and fresh alfalfa, and occasionally, they were given branches from trees such as palms, ficus and acacias. Water and blocks of mineral salt and oligoelements were available ad libitum.

### 2.2. Distributions of Conceptions Dates throughout the Year

Dates of birth for animals kept at EEZA of the three species of gazelles, from 1995 until 2005, were taken from studbooks [5,6,7]. Average gestation period in Cuvier’s gazelle is 161 days [10], 169.4 days in dorcas gazelle [42] and 195 days in Mohor gazelle [7]. To calculate conception date, the number of days corresponding to the average gestation period were subtracted from the date of birth.

### 2.3. Effect of Seasonality

#### 2.3.1. Animals and Body Measures

Four adult males (ages ranging from 3 years, 9 months, to 8 years) per species housed alone were sampled to assess seasonal effects. Animals were weighed and testicular length and width were measured (cm) with calipers through the scrotal sac during anesthesia and before electroejaculation. The weight of each testis was calculated from dimensions as described by Harcourt et al. [43]. Testicular volume was calculated by the ellipsoid formula = 4/3 × π × L/2 × B1/2 × B2/2, where L is the length and B1 and B2 are two breadths of the ellipsoid. We assumed both breadths to be equal. Testicular weights were calculated by the following formula: weight (g) = volume (cm^3^) × 1.1, where 1.1 was considered the tissue density. Left and right testes weights were added, and relative testes weight was calculated by dividing testes weight by body weight.

#### 2.3.2. Hormone Analyses

Blood samples were taken from the jugular vein in anesthetized animals. Samples were collected in heparin-containing tubules. Plasma was recovered by centrifugation, frozen at −20 °C and then stored at −80 °C until assayed. Testosterone was determined in duplicate by radioimmunoassay (Laboratorie de Dosages Hormonaux, INRA, Tours, France).

#### 2.3.3. Semen Collection and Evaluation

Semen was obtained by electroejaculation throughout 12 months. Collection of semen was performed every two months starting in December 2000 for Cuvier’s gazelle, February 2001 for dorcas gazelle and April 2001 for Mohor gazelle. One male of Cuvier’s gazelle died in May 2001 and one dorcas gazelle male had to be removed from the study in November 2001.

Semen was collected by electroejaculation under surgical anesthesia, and stimulating voltage and probe dimensions for the three species were as described previously [39,41]. Semen was placed at 30 °C in a water bath pending analyses, which were performed shortly after collection.

The methods used to evaluate the semen samples have been described previously [30,39]. Briefly, semen volume (mL) and sperm concentration (cells/mL) were measured, and total number of spermatozoa was calculated by multiplying both parameters. Also, wave motion was assessed at 100× magnification as a subjective score from 0 to 5 and, subsequently, semen aliquots were diluted in Phosphate-buffered saline with bovine serum albumin (5 mg/mL) and used to evaluate individual and progressively motile sperm and quality of motility, that was assessed using a scale of 0 (lowest) to 5 (highest) at 400× magnification on a Nikon E400 microscope. For sperm assessments, a minimum of 100 spermatozoa were counted in each preparation. A Sperm Motility Index (SMI) was calculated as: SMI = (% Individual motility + (Quality of motility × 20)) × 0.5. Moreover, semen aliquots were evaluated for acrosomal status and viability with eosin-nigrosin and Giemsa, as described previously [39,41].

### 2.4. Effect of Housing Conditions

Male gazelles housed alone (11 males for Cuvier’s, 7 for dorcas and 11 for Mohor gazelles), in groups of males (3 for Cuvier’s, 10 for dorcas and 11 for Mohor gazelles) or with a group of females (3 for Cuvier’s, 4 for dorcas and 4 for Mohor gazelles) were anesthetized and electro-ejaculated between October and December 1996 and October and November 1997, as described previously. Testes weights and relative testes weights were quantified, and semen was assessed as described above. Blood samples were collected, and testosterone and cortisol were analyzed as described above.

### 2.5. Statistical Analysis

Statistical analyses were performed with IBM SPSS 26 for Windows (SPSS Inc., Chicago, IL, USA). Conception dates were analyzed by Chi-Square. Normality was tested with the Shapiro–Wilk test and data were transformed (arcsin for percentages and log10 for other variables) when needed. To analyze the relation between conceptions and environmental variables, a Pearson correlation analysis was performed. Linear Mixed Model was used with Bonferroni adjustment to compare seminal parameters, body measures and hormones levels throughout the year. Differences due to housing conditions were analyzed by the General Linear Model (GLM) using the age in days as a covariable and the Bonferroni correction as a pairwise comparison. When data could not be normalized, the Kruskal–Wallis test was used. Data are expressed as means ± standard error of the mean (SEM) and *p* < 0.05 was considered statistically significant.

## 3. Results

### 3.1. Distributions of Conception Dates throughout the Year

The three species conceived in every month of the year, but all three species showed statistical differences compared to a homogeneous distribution of conceptions throughout the year (Cuvier’s, χ^2^ = 40.2, *p* < 0.001, dorcas, χ^2^ = 21.9, *p* = 0.025 and Mohor, χ^2^ = 25.2, *p* = 0.009). Cuvier’s gazelle presented a bimodal distribution, with 30% of conceptions occurring between February and April (mainly in April with 14%) and 40% between October and December (Figure 2a). In dorcas gazelle, no clear seasonal pattern was present, but conceptions increased in April and October (Figure 2b). Mohor gazelle had a peak of conceptions between July and October, when nearly 48% of conceptions took place (Figure 2c). Percentage of conceptions in Cuvier’s gazelle was negatively correlated with photoperiod (*p* = 0.017, r^2^ = 0.484) and positively correlated with rainfall (*p* = 0.036, r^2^ = 0.37), while conceptions in Mohor gazelle was positively correlated with temperature (*p* = 0.042, r^2^ = 0.353). No relations were found among conceptions in dorcas gazelle and environmental variables.

### 3.2. Effect of Seasonality

#### 3.2.1. Body Measures

The three species of gazelles differ in size (Table 1). Mohor gazelle is larger than Cuvier’s which, in turn, is larger than dorcas (*p* < 0.001). Testes weight was higher in Mohor gazelles (*p* ≤ 0.003) but no differences were found between Cuvier’s and dorcas gazelles (Table 1). Consequently, relative testes size was statistically different among species, being higher in dorcas, intermediate in Cuvier’s and lower in Mohor gazelle (*p* < 0.001; Table 1). When comparing between months, there were no variations in body weight, testes size and relative testes size in Cuvier’s and dorcas gazelles. On the other hand, when testes size was analyzed in Mohor gazelle, there were no differences in the comparison among months, although the model showed an effect of seasonality (*p* = 0.049). Mohor gazelle relative testes size was lower (*p* ≤ 0.003) in December than in June, August and October (0.0068 vs. 0.0087–0.0091).

#### 3.2.2. Testosterone Levels

Overall, average serum testosterone levels in Mohor gazelle (1.4 ± 0.2 ng/mL) were lower than those in Cuvier’s (4.3 ± 0.6 ng/mL) or dorcas gazelles (4.4 ± 0.5 ng/mL; *p* < 0.001), and they remained lower than those in the other two species throughout the year. Testosterone levels in Cuvier’s (*p* = 0.046) and Mohor gazelles (*p* = 0.030) were affected by season, but the pair-wise comparison did not show a statistical difference between months (Figure 3). Testosterone levels in dorcas gazelle followed a similar pattern to those in Cuvier’s but the model showed no effect of season.

#### 3.2.3. Seminal Parameters

In Cuvier’s gazelle, the total number of sperm was higher in April than in December (*p* = 0.022) or in June (*p* = 0.037; Figure 4e). None of the other sperm parameters were affected by season.

Quality of sperm parameters in dorcas gazelle showed a drop in October (Figure 4). The model showed a variation throughout the year in the percentage of progressive sperm, quality of motility, SMI and viability (*p* ≤ 0.047). When comparing among months, statistically lower progressive sperm and SMI were found in October than in August (*p* ≤ 0.041).

Mohor gazelle quality of motility and viability showed variation throughout the year (*p* ≤ 0.037), but no differences were found in the comparison among months (Figure 4).

### 3.3. Effect of Housing Conditions

#### 3.3.1. Body Measures

There were no differences in body, testes and relative testes weights in Cuvier’s, Mohor or dorcas gazelles associated with housing conditions.

#### 3.3.2. Testosterone and Cortisol Levels

Testosterone and cortisol levels did not show differences (*p* > 0.05) within species under different housing conditions (Figure 5). The model showed that cortisol in Mohor gazelles was higher in males kept with females, but this effect was due to age. No differences (*p* > 0.05) in testosterone and cortisol level were found between young (less than 2.5 years) and adult gazelles for the three species.

#### 3.3.3. Seminal Parameters

In Cuvier’s and Mohor gazelles, semen parameters were not affected by housing (Table 2). Males of dorcas gazelle kept alone presented better sperm quality than those housed with females. Lower sperm concentration, total number of spermatozoa, wave motion, individual motility, progressive motility and quality of motility were observed in dorcas males housed with females than in males kept individually (*p* ≤ 0.037). In addition, males housed with females presented lower sperm concentration and total number of sperm than males housed in bachelor groups (*p* ≤ 0.044). Finally, dorcas males housed alone had higher quality of motility than males housed with other males (*p* = 0.033). When testes weight was added to the model, the differences in progressive motility due to housing disappeared.

## 4. Discussion

The results of this study suggest that there is a seasonal pattern in conception dates, with semen parameters also showing seasonal variation in these three gazelle species, although with different intensity. In addition, housing conditions affected the three species differently, so that while in dorcas gazelle, solitary males showed better semen quality than males kept with females, in Cuvier’s and Mohor gazelles, analyzed semen parameters were not affected by housing conditions.

Few studies have been performed with wild bovids on sperm seasonality. In any case, they have revealed both species-specific differences and variations between captive and free-living males [19,20,21,22,23] and variations in testicular function and the quality of semen throughout the year [19,21,24,25,26,28,38]. Cuvier’s gazelle semen showed higher total number of sperm in April associated with a period of higher conception rate. However, sperm parameters were lower during October–December when a second peak in conceptions was observed. A female effect could be associated with increased conception periods found in this study. Males of Mohor gazelle have been shown to be able to detect the approach of estrous [44]. A seasonal anestrus have been found in North America captive Addra gazelle although births occurred throughout the year [15]. On the other hand, testosterone levels increased in August preceding this rise in conception rates, as was found in Arabian sand gazelle (*Gazella subgutrosa marica*) [29]. It could be proposed that behavior plays an important role at this stage for conception rates and more aggressive activities are displayed by males in response to higher testosterone levels that could be associated with higher mating rate.

Previous studies in Cuvier’s gazelle kept at the EEZA in the early years of the captive breeding program showed that conceptions were almost exclusively concentrated from September to November [6,9]. Similarly, in the wild, births in Cuvier’s gazelle have been described to occur in April and May [4]. Our data indicate that conceptions also occur in April in the captive population, with subsequent births in September. Conceptions in this species were positively related with rainfall. This seems to be an adaptation to the new environment in Almeria, where two periods of rain exist. Although these animals are confined to a captive breeding facility, with daily access to food, and give birth all year around, higher conceptions rates are maintained in periods that would lead to births when rain is maximal.

Dorcas gazelles’ higher conception rates occurred in October, and thus a higher prevalence of parturitions was seen in April and May, as was previously described in this species [5]. There was also another slight increase in conception rates in April, although it was not reflected in a second peak of parturitions. However, sperm quality was lower in October, suggesting that conception rate may increase by other mechanism. Although testosterone level did not vary significantly, a rise was seen in August and could be leading to an increase in conceptions by behavioral changes or a female effect, as was proposed for Cuvier’s gazelle. Captive dorcas gazelles kept at a similar latitude (38°51′ N) in the United States experienced no seasonal variations in semen quality [20], but this is at variance with our results showing a drop in semen quality in October. On the other hand, testosterone levels were found not to vary throughout the year [20], a result confirmed in this study. Overall, the distribution of conceptions in this species could not be explained by male semen characteristics and was not related with environment data.

The distribution of conceptions in Mohor gazelle, with a main rise in the period between July and October, is in agreement with previous reports in the same captive center [14]. Parturitions in EEZA were mainly distributed from February to April, corresponding with a period of higher rainfall, and a second smaller peak in parturitions was presented in September corresponding with a second rainfall period [14]. It has been proposed that Mohor gazelles adjust their parturition period to the raining season, as observed for Mohor gazelles translocated from Almeria to Senegal [14]. Similarly, births in Addra gazelle kept in captivity in North America occurred throughout the year but a peak of births was observed between July and September that coincides with maximal rainfall in North America and in historical African range [15]. In our study, sperm viability and quality of motility in Mohor gazelle were higher in April and August. Similarly, testosterone level increased in August and relative testes size was higher from June to October. Seasonality in fecal metabolites of testosterone was also found in Mohor gazelle kept at the EEZA [45]. Better sperm quality in August, higher testosterone level and relative testes size explain the highest peak of conception in that period that was also related with higher temperatures but not with rainfall.

It seems that captive conditions have allowed for an extension of the parturition period that could take place all over the year in the three species. But they maintain a higher parturition rate in the period of higher rainfall: from March to May and September for Cuvier’s gazelle, from April to May in dorcas and from February to April in Mohor gazelle.

Testosterone levels throughout the year in Mohor gazelle were considerably lower than those in Cuvier’s and dorcas gazelles due to species-specific differences. However, relatively higher testosterone levels in Mohor gazelle were found in the experiment of housing condition. Both experiments were not performed in the same year and the distribution of testosterone peaks was probably different. Variability in baseline testosterone levels and peak values between species was also observed in other wild and domestic bovid species [25,46,47,48].

When the effect of housing was analyzed in Cuvier’s and Mohor gazelles, semen parameters were not affected, suggesting that sharing the enclosure with other individuals did not present an effect over the semen quality in these two species. Interestingly, most sperm parameters in dorcas gazelle males were higher in males kept alone and lower in males sharing the enclosure with females. Males housed with females could be undergoing a process of sperm depletion due to frequent mating with females compared with single housed males [49,50,51]. This constrained mature sperm availability and higher activity rate in males kept with females could explain the differences in semen quality. Also, males kept individually presented higher quality of motility than males housed in bachelor groups, showing some impact of social life over sperm quality. On the other hand, at the EEZA captive breeding center, Mohor and dorcas males kept in female herds may share the enclosures, whereas Cuvier’s gazelle males are kept in single species groups to avoid aggression. It could be proposed that sharing the enclosures with another species increases activity levels and that these could be negatively affecting the ejaculate of dorcas gazelle, the smallest species. No effect of housing was found on cortisol level in dorcas gazelle, while, in other study, more aggressive behaviors were found in male groups compared with female groups with younglings that positively correlate with hair cortisol [52]. Besides, it has to be taken into account that analysis of spermatozoa for this experiment was performed in October, November and December. Results from the first experiment showed that sperm quality was lower in October for dorcas gazelle and could be maximizing the effect of housing. In contrast, Cuvier’s and Mohor gazelle sperm parameters were not affected during this period. Therefore, the interaction between seasonality and housing was different for these species. Further studies will be required to verify whether the negative effect of social housing in dorcas gazelle is maintained in periods of higher sperm quality.

## 5. Conclusions

Semen quality was affected by season in Almeria in these three species of gazelles, with better ejaculate quality in April for Cuvier’s gazelle, and April and August for Mohor gazelle, whereas dorcas gazelle exhibited worse semen parameters in October. Hence, semen collection and cryopreservation for banking could be concentrated in the months with better semen quality. Ejaculate parameters in dorcas gazelle males housed singly were better than those in males sharing the enclosure with females. Consequently, it is advisable that collection of semen be performed in single housed males in this species. On the other hand, ejaculate quality in Cuvier’s and Mohor gazelles was not affected by housing. Taken together, the results of this study indicate that the modulation of semen quality caused by environmental and social factors affects these three gazelle species differently, and that this will be of great importance for collection and preservation of male gametes for genome resource banks of these species.

## Figures and Tables

**Figure 1 animals-11-00901-f001:**
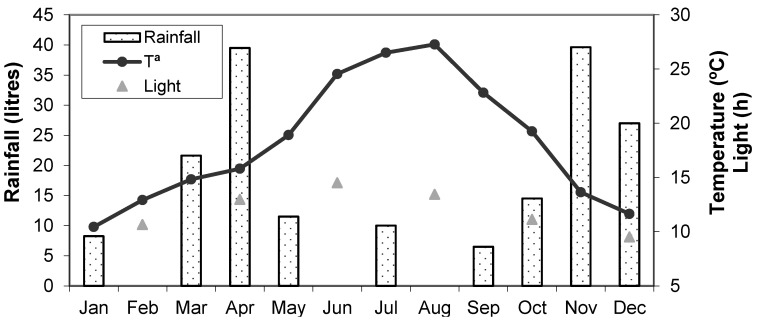
Environmental data in Almeria. Average monthly temperature from 2000 to 2001, rainfall from 2001 to 2002 and photoperiod from 2001, data obtained from Plataforma Solar de Almeria.

**Figure 2 animals-11-00901-f002:**
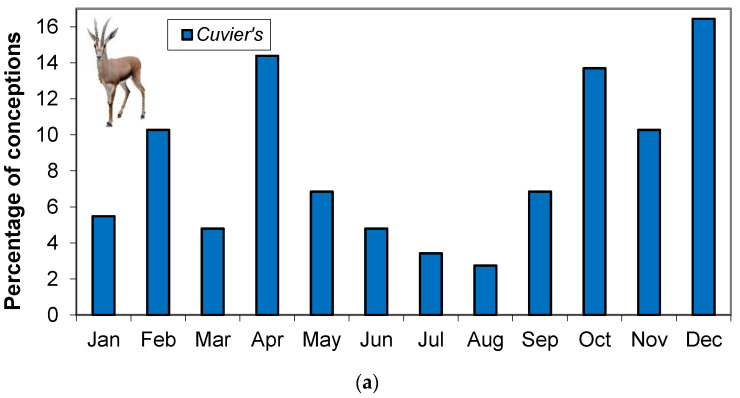
Distribution of conception dates for (**a**) Cuvier’s gazelle, (**b**) dorcas gazelle and (**c**) Mohor gazelle, data of dates of birth at Estación Experimental de Zonas Áridas from January 1995 to December 2005 were taken from studbooks. Cuvier’s gazelle *n* = 146, dorcas gazelle *n* = 192 and Mohor gazelle *n* = 217.

**Figure 3 animals-11-00901-f003:**
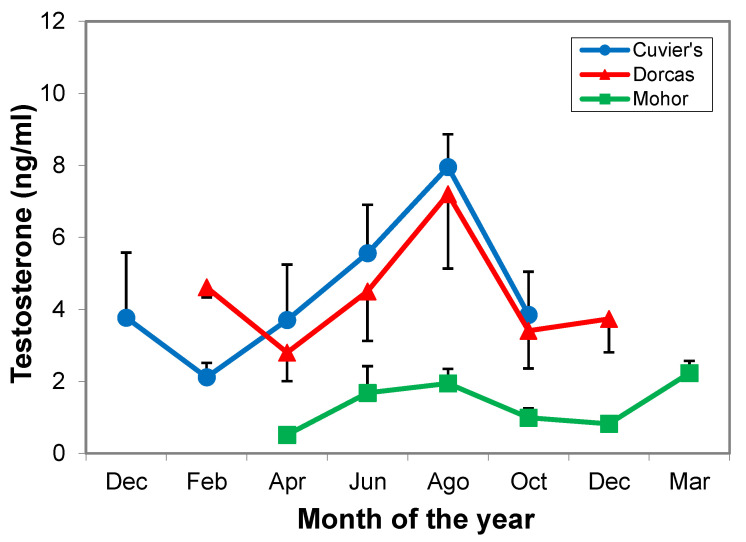
Testosterone levels in three species of gazelles from December 2000 to March 2002. *n* = 4 males per species.

**Figure 4 animals-11-00901-f004:**
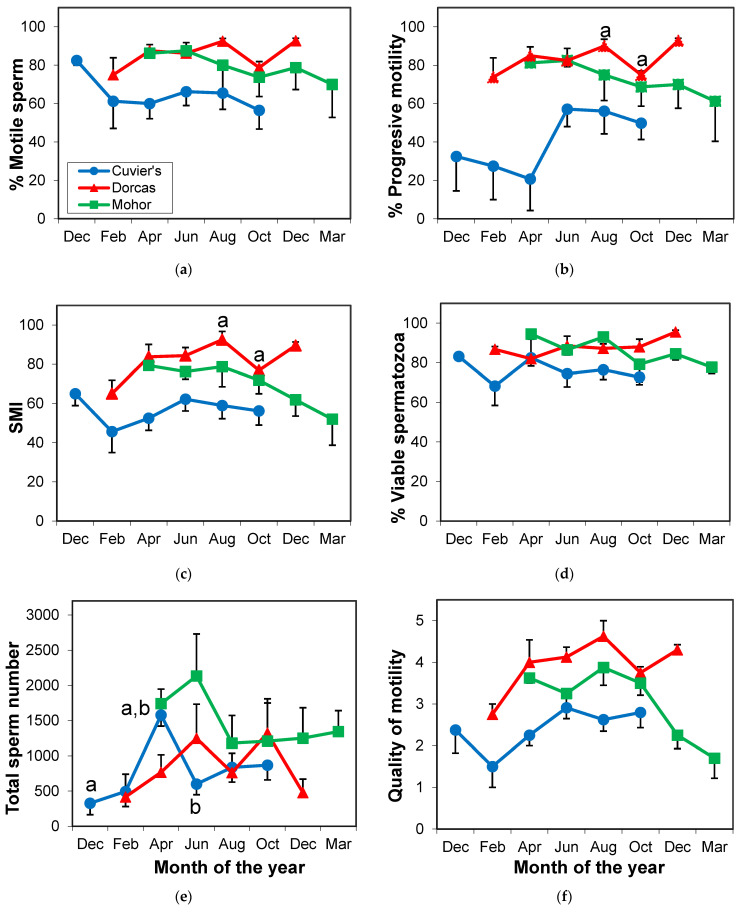
Variation in sperm parameters throughout the year in Cuvier’s, dorcas and Mohor gazelles in Almeria. (**a**) Percentage of individual motility, (**b**) percentage of progressive sperm, (**c**) sperm motility index (SMI), (**d**) percentage of viable spermatozoa, (**e**) total number of spermatozoa and (**f**) quality of motility. Similar letters indicate differences between months. *n* = 4 males/species.

**Figure 5 animals-11-00901-f005:**
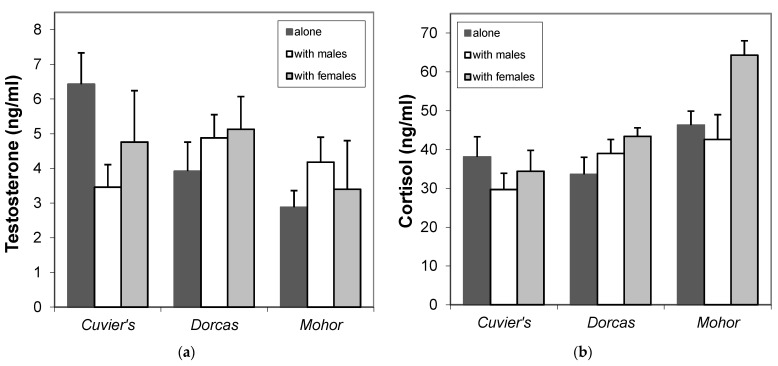
Hormone levels in males of three gazelle species kept alone, with males or with females sampled in October–December 1996 and October–November 1997. (**a**) Testosterone and (**b**) cortisol levels. Cuvier’s gazelles *n* = 11, 3 and 3, dorcas gazelles *n* = 7, 10 and 4, Mohor gazelles *n* = 11, 11 and 4 for males housed alone, with other males and with females, respectively.

**Table 1 animals-11-00901-t001:** Body, testes and relative testes weights in three species of gazelles.

	Cuvier’s Gazelle	Dorcas Gazelle	Mohor Gazelle
Body weight (kg)	34.2 ± 0.8 ^a^	17.2 ± 0.6 ^b^	63.3 ± 1.3 ^c^
Testes weight (g)	47.0 ± 3.6 ^a,b^	40.8 ± 4.1 ^a^	54.4 ± 1.4 ^b^
Relative testes weight (g/kg)	1.38 ± 0.11 ^a^	2.39 ± 0.28 ^b^	0.86 ± 0.01 ^c^

^a,b,c^ Different letters indicate statistically significant differences between species. Cuvier’s gazelle *n* = 4, Mohor gazelle *n* = 4 and dorcas gazelle *n* = 4. For each male, the mean among data of the six measures was used. Mean ± standard error of the mean (SEM).

**Table 2 animals-11-00901-t002:** Semen parameters in males of three species of gazelles housed alone, with males or with females.

	Cuvier´s Gazelle	Dorcas Gazelle	Mohor Gazelle
**Volume (µL)**			
Alone	626.8 ± 148.7	717.0 ± 209.9	937.0 ± 61.1
with males	1019.0 ± 288.2	392.1 ± 118.1	1566.4 ± 486.1
with females	294.7 ± 91.0	61.5 ± 26.4	2162.5 ± 589.0
**Sperm concentration (×10^6^/mL)**			
Alone	649.3 ± 176.2	1475.2 ± 436.3 ^a^	789.5 ± 90.4
with males	559.3 ± 11.2	697.9 ± 233.2 ^b^	564.4 ±186.6
with females	246.7 ± 168.2	99.6 ± 71.1 ^a,b^	329.1 ± 84.3
**Total sperm number (×10^6^)**			
Alone	547.4 ± 283.2	679.5 ± 137.1 ^a^	754.4 ± 109.1
with males	576.2 ± 169.3	319.0 ± 117.0 ^b^	961.6 ± 340.2
with females	103.2 ± 84.5	11.7 ± 10.5 ^a,b^	712.1 ± 261.0
**Wave motion (scale 0–5)**			
Alone	1.9 ± 0.5	4.8 ± 0.2 ^a^	3.7 ± 0.3
with males	2.3 ± 1.2	2.8 ± 0.6	2.4 ± 0.6
with females	1.0 ± 1.0	0.5 ± 0.3 ^a^	2.5 ± 0.7
**Individual motility (%)**			
Alone	49.5 ± 10.6	92.2 ± 2.9 ^a^	87.4 ± 4.0
with males	64.0 ± 29.5	55.8 ± 12.4	72.3 ± 10.3
with females	28.6 ± 28.6	26.0 ± 16.8 ^a^	78.1 ± 16.9
**Progressive motility (%)**			
Alone	32.9 ± 10.9	88.2 ± 2.8 ^a^	75.1 ± 5.7
with males	25.3 ± 19.3	45.8 ± 12.8	47.3 ± 10.4
with females	23.8 ± 23.8	26.0 ± 16.8 ^a^	68.0 ± 13.2
**Quality of motility (scale 0–5)**			
Alone	2.1 ± 0.5	4.8 ± 0.2 ^a^	3.5 ± 0.2
with males	2.0 ± 1.0	2.9 ± 0.7^b^	2.3 ± 0.4
with females	1.0 ± 1.0	2.0 ± 1.2 ^a,b^	3.5 ± 0.3
**Viability (%)**			
Alone	56.4 ± 8.3	91.0 ± 2.2	85.4 ± 2.1
with males	77.0 ± 4.4	73.4 ± 11.2	72.1 ± 6.9
with females	30.0 ± 19.6	55.7 ± 17.4	71.8 ± 7.4
**Intact acrosomes (%)**			
Alone	66.2 ± 7.0 ^a^	93.4 ± 1.5	88.1 ± 2.4
with males	95.2 ± 0.6 ^b^	90.3 ± 3.4	78.4 ± 10.2
with females	66.1 ± 17.0	84.5 ± 2.5	86.9 ± 6.3

^a,b^ Different letters indicate statistical differences between males housed alone, with other males or with females. Cuvier’s gazelles *n* = 11, 3 and 3, dorcas gazelles *n* = 7, 10 and 4, and for Mohor gazelles, *n* = 10, 10 and 4 for animals housed alone, with males and with females, respectively. Mean ± SEM.

## Data Availability

Data will be made available on request.

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
