# Peer review of "Effect of Season and Social Environment on Semen Quality and Endocrine Profiles of Three Endangered Ungulates (Gazella cuvieri, G. dorcas and Nanger dama)"

_animals, 2021, doi:10.3390/ani11030901_

Round 1

Reviewer 1 Report

The Authors have addressed all my concerns and I have no further comments. 

As far as I am concerned, the manuscript is now acceptable to be published.

(In the figures some panels appear duplicated, but perhaps it is a problem of the track changes mode)

Reviewer 2 Report

The revised MS contains all comments adequately addressed.

This manuscript is a resubmission of an earlier submission. The following is a list of the peer review reports and author responses from that submission.

Round 1

Reviewer 1 Report

The manuscript entitled “Effect of season and social environment on semen quality and endocrine profiles of three endangered ungulates (Gazella cuvieri, G. dorcas and Nanger dama)” has been carefully evaluated. In this study, the authors provide data about sperm parameters and males of three different species kept in a zoo. Although the study provides some interesting new data about ejaculate, sperm viability, and testes’ conformation, there are too many weak points. Unfortunately, the study claims to investigate the influence of season on different male parameters, but the female part is completely neglected. It would be interesting to know if there are also changes in the activity of the ovary with regards to the season what would explain the conception rate in different months much clearer. Another weak point is the missing data from wildlife counterparts, to compare if the described parameters are perhaps influenced by the housing condition in zoos. Moreover, the seasons and climate in Almeria are not a proper model for those ones occurring in the natural habitat of the three named species, so one has to be very precise with conclusions. The study lacks data that could emphasize in a better way difference of sperm viability (i.e. CASA) or the conformation of testes ( Ultrasound examination). Moreover, there is no statement about the “Ethical Approval” of the made investigations (the harm of electro-ejaculation might be an issue). All these points are open questions and missing in the manuscript but they are mandatory to clearly provide a conclusion if the season or housing conditions might have an influence on male reproduction of three ungulate species kept in zoos. 

Please note my specific comments below.

L13/14: This does play a role not only in endangered species but also in many other domestic species. Please fix.

L48: Although IVF is a very common abbreviation, please introduce it first before using it.

L53: Wouldn’t it be better to use the phrase “Mhorr gazelle” instead of “Mohor” since it is closer to the Latin one?

L60: I think there is a typo in “thought”- shouldn’t it be “through” ?

L61: Please provide here the mean pregnancy duration for the reader.

L69: For sure the accumulation of birth is also due to female factors, so please be more precise with your statements.

L86: Please replace “10” with “ten”.

L87-89: Maybe the reason why the effect of season has never been described before is due to the fact that there is not really a “season” in their wildlife habitats? Therefore would be no reason to investigate?

L91-95: Please provide other species as examples that are closer to your three investigated species, i.e. deers et cetera.

L107: Objective (1) is too unprecise, should be added “kept in European zoos” to inform the reader that no wildlife conditions are investigated, and no southern hemisphere conditions.

L147: For which species the RIA was dedicated for ?

L168: How was the “wave-motion” assessed ? There a no CASA data, please explain why this useful tool has not been used.

Figure 2 a-c: The means are provided without SEM or SD, please explain why or fix.

Table 1: Do the values show the mean of one time point and different individuals, or the mean of different time points and different individuals? Please fix the title and legend.

L230: Did you measure free testosterone?

L384: Did you check the temperature 2-3 months before October? Maybe there was enormous heat stress which influenced the spermatogenesis in this species and which was then influencing the results in October?

Reviewer 2 Report

The aim of this paper was essentially to explore the effect of seasonality and social environment on sperm quality in three endangered gazelles.

The topic of the study is interesting, because the knowledge of factors affecting semen quality could be of great importance for the collection and preservation of semen from threatened animals.

However, results are mainly descriptive.

There are several limitations of this study which should be mentioned:

  • Lines 155-158: When did the Authors perform sperm evaluation? About ten years ago? Why weren’t the results published earlier?
  • Possible suggested physiological mechanisms for the observed effects are not investigated.
  • Limitations of this study should be discussed.

Minor:

Figure 1: Names of some months are cut (Apr, May, Aug, Sep)

Lines 150-152: Collection of semen was performed every two months starting in… until…

Table 1: Relative testes weight (g/kg)

Figure 4: The legend with the colors of the curves is missing